# Playing hard exploration games by watching YouTube

**Yusuf Aytar**[*]**, Tobias Pfaff**[*]**, David Budden, Tom Le Paine, Ziyu Wang, Nando de Freitas**

DeepMind, London, UK
{yusufaytar,tpfaff,budden,tpaine,ziyu,nandodefreitas}@google.com

## Abstract

Deep reinforcement learning methods traditionally struggle with tasks where environment rewards are particularly sparse. One successful method of guiding exploration in these domains is to imitate trajectories provided by a human demonstrator. However, these demonstrations are typically collected under artificial conditions, i.e. with access to the agent's exact environment setup and the demonstrator's action and reward trajectories. Here we propose a two-stage method that overcomes these limitations by relying on noisy, unaligned footage without access to such data. First, we learn to map unaligned videos from multiple sources to a common representation using self-supervised objectives constructed over both time and modality (i.e. vision and sound). Second, we embed a single YouTube video in this representation to construct a reward function that encourages an agent to imitate human gameplay. This method of one-shot imitation allows our agent to convincingly exceed human-level performance on the infamously hard exploration games MONTEZUMA'S REVENGE, PITFALL! and PRIVATE EYE for the first time, even if the agent is not presented with any environment rewards.

## 1   Introduction

People learn many tasks, from knitting to dancing to playing games, by watching videos online. They demonstrate a remarkable ability to transfer knowledge from the online demonstrations to the task at hand, despite huge gaps in timing, visual appearance, sensing modalities, and body differences. This rich setup with abundant unlabeled data motivates a research agenda in AI, which could result in significant progress in third-person imitation, self-supervised learning, reinforcement learning (RL) and related areas. In this paper, we show how this proposed research agenda enables us to make some initial progress in self-supervised alignment of noisy demonstration sequences for RL agents, enabling human-level performance on the most complex and previously unsolved Atari 2600 games.

Despite the recent advancements in deep reinforcement learning algorithms [7, 9, 17, 19] and architectures [22, 35], there are many "hard exploration" challenges, characterized by particularly sparse environment rewards, that continue to pose a difficult challenge for existing RL agents. One epitomizing example is Atari's MONTEZUMA'S REVENGE [10], which requires a human-like avatar to navigate a series of platforms and obstacles (the nature of which change substantially room-to-room) to collect point-scoring items. Such tasks are practically impossible using naive $\epsilon$-greedy exploration methods, as the number of possible action trajectories grows exponentially in the number of frames separating rewards. For example, reaching the first environment reward in MONTEZUMA'S REVENGE takes approximately 100 environment steps, equivalent to $100^{18}$ possible action sequences. Even if a reward is randomly encountered, $\gamma$-discounted RL struggles to learn stably if this signal is backed-up across particularly long time horizons.

---

[*]denotes equal contribution

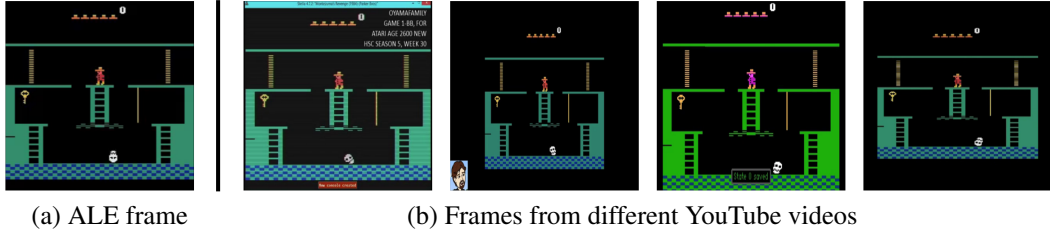

(a) ALE frame                    (b) Frames from different YouTube videos

Figure 1: Illustration of the domain gap that exists between the Arcade Learning Environment and YouTube videos from which our agent is able to learn to play MONTEZUMA'S REVENGE. Note different size, resolution, aspect ratio, color and addition of visual artifacts such as text and avatars.

Successful attempts at overcoming the issue of sparse rewards have fallen broadly into two categories of guided exploration. First, intrinsic motivation methods provide an auxiliary reward that encourages the agent to explore states or action trajectories that are "novel" or "informative" with respect to some measure [8, 30, 27]. These methods tend to help agents to re-explore discovered parts of state space that appear novel or uncertain (known-unknowns), but often fail to provide guidance about where in the environment such states are to be found in the first place (unknown-unknowns). Accordingly, these methods typically rely on an additional random component to drive the initial exploration process. The other category is imitation learning [20, 41], whereby a human demonstrator generates state-action trajectories that are used to guide exploration toward areas considered salient with respect to their inductive biases. These biases prove to be a very useful constraint in the context of Atari, as humans can immediately identify e.g. that a skull represents danger, or that a key unlocks a door.

Among existing imitation learning methods, DQfD by Hester et al. [20] has shown the best performance on Atari's hardest exploration games. Despite these impressive results, there are two limitations of DQfD [20] and related methods. First, they assume that there is no "domain gap" between the agent's and demonstrator's observation space, e.g. variations in color or resolution, or the introduction of other visual artifacts. An example of domain gap in MONTEZUMA'S REVENGE is shown in Figure 1, considering the first frame of (a) our environment compared to (b) YouTube gameplay footage. Second, they assume that the agent has access to the exact action and reward sequences that led to the demonstrator's observation trajectory. In both cases, these assumptions constrain the set of useful demonstrations to those collected under artificial conditions, typically requiring a specialized software stack for the sole purpose of RL agent training.

To address these limitations, this paper proposes a method for overcoming domain gaps between the observation sequences of multiple demonstrations, by using self-supervised classification tasks that are constructed over both time (temporal distance classification) and modality (cross-modal temporal distance classification) to learn a common representation (see Figure 2). Unlike previous approaches, our method requires neither (a) frame-by-frame alignment between demonstrations, or (b) class labels or other annotations from which an alignment might be indirectly inferred. We additionally propose a new unsupervised measure (cycle-consistency) for evaluating the quality of such a learnt embedding.

Using our embedding, we propose an auxiliary imitation loss that allows an agent to successfully play hard exploration games without requiring the knowledge of the demonstrator's action trajectory. Specifically, providing a standard RL agent with an imitation reward learnt from a single YouTube video, we are the first to convincingly exceed human-level performance on three of Atari's hardest exploration games: MONTEZUMA'S REVENGE, PITFALL! and PRIVATE EYE. Despite the challenges of designing reward functions [18, 36] or learning them using inverse reinforcement learning [1, 49], we also achieve human-level performance even in the absence of an environment reward signal.

## 2   Related Work

Imitation learning methods such as DQfD have yielded promising results for guiding agent exploration in sparse-reward tasks, both in game-playing [20, 32] and robotics domains [41]. However, these methods have traditionally leveraged observations collected in artificial conditions, i.e. in the absence of a domain gap (see Figure 1) and with full visibility over the demonstrator's action and reward trajectories. Other approaches include interacting with the environment before introducing the expert

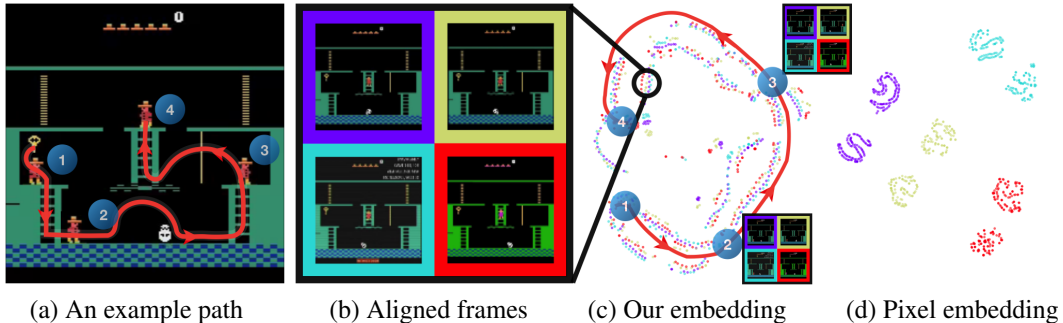

|                  |                    |                    |                       |
|------------------|--------------------|--------------------|-----------------------|
| (a) An example path | (b) Aligned frames | (c) Our embedding | (d) Pixel embedding |

Figure 2: For the path shown in (a), t-SNE projections [25] of observation sequences using (c) our embedding, versus (d) raw pixels. Four different domains are compared side-by-side in (b) for an example frame of MONTEZUMA'S REVENGE: (purple) the Arcade Learning Environment, (cyan/yellow) two YouTube training videos, and (red) an unobserved YouTube video. It is evident that all four trajectories are well-aligned in our embedding space, despite (purple) and (red) being held-aside during training. Using raw pixel values fails to achieve any meaningful alignment.

demonstrations [38, 31] and goal conditioned policies for high fidelity imitation [29], although these papers typically do not assume domain gap or operate in sparse reward settings.

There are several methods of overcoming the domain gap in the previous literature. In the simple scenario of demonstrations that are aligned frame-by-frame [24, 34], methods such as CCA [2], DCTW [39] or time-contrastive networks (TCN) [34] can be used to learn a common representation space. However, YouTube videos of Atari gameplay are more complex, as the actions taken by different demonstrators can lead to very different observation sequences lacking such an alignment. In this scenario, another common approach for domain alignment involves solving a shared auxiliary objective across the domains [5, 6]. For example, Aytar et al. [5] demonstrated that by solving the same scene classification task bottlenecked by a common decision mechanism (i.e. using the same network), several very different domains (i.e. natural images, line drawings and text descriptions) could be successfully aligned. Similarly, domain adaptive meta-learning [44] uses a shared policy network for addressing the domain gap for robotic tasks, though they require both first person robot demonstrations and third person human demonstrations. Our work differs from the above approaches in that we do not make use of any category-guided supervision or first person demonstrations. Instead, we define our shared tasks using self-supervision over unlabeled data. This idea is motivated by several recent works in the self-supervised feature learning literature [3, 11, 12, 28, 42, 45, 26].

Other related approaches include single-view TCN [34], which is another self-supervised task that does not require paired training data. We differ from this work by using temporal classification instead of triplet-based ranking, which removes the need to empirically tune sensitive hyper parameters (local neighborhood size, ranking margin, etc). Another approach [33] performs temporal classification but limits its categories to frames close or far away in time. With respect to our use of cross-modal data, another similar existing method in the feature learning literature is $L^3$-net [3]. This approach learns to align vision and sound modalities, whereas we learn to align multiple audio-visual sequences (i.e. demonstrations) using multi-modal alignment as a self-supervised objective. We adapt both TCN and $L^3$-net for domain alignment and provide an evaluation compared to our proposed method in Section 6. We also experimented with third-person imitation methods [37] that combine the ideas of generative adversarial imitation learning (GAIL) [21] and adversarial domain confusion [16, 40], but were unable to make progress using the very long YouTube demonstration trajectories.

Considering the imitation component of our work, one perspective is that we are learning a reward function that explains the demonstrator's behavior, which is closely related to inverse reinforcement learning [1, 49]. There have also been many previous studies that consider supervised [4] and few-shot methods [13, 15] for imitation learning. However, in both cases, our setting is more complex due to the presence of domain gap and absence of demonstrator action and reward sequences.

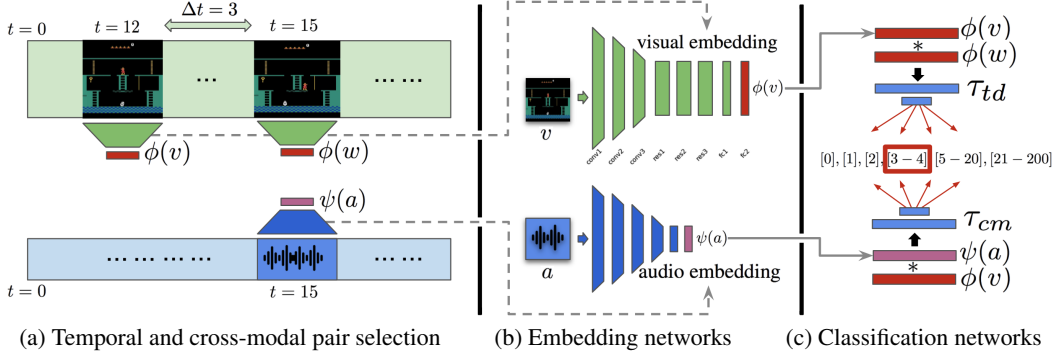

(a) Temporal and cross-modal pair selection     (b) Embedding networks     (c) Classification networks

Figure 3: Illustration of the network architectures and interactions involved in our combined TDC+CMC self-supervised loss calculation. The final layer FC2 of $\phi$ is later used to embed the demonstration trajectory to imitate. Although the Arcade Learning Environment does not expose an audio signal to our agent at training time, the audio signal present in YouTube footage made a substantial contribution to the learnt visual embedding function $\phi$.

## 3 Closing the domain gap

Learning from YouTube videos is made difficult by both the lack of frame-by-frame alignment, and the presence of domain-specific variations in color, resolution, screen alignment and other visual artifacts. We propose that by learning a common representation across multiple demonstrations, our method will generalize to agent observations without ever being explicitly exposed to the Atari environment. In the absence of pre-aligned data, we adopt self-supervision in order to learn this embedding. The rationale of self-supervision is to propose an auxiliary task that we learn to solve simultaneously across all domains, thus encouraging the network to learn a common representation. This is motivated by the work of Aytar et al. [5], but differs in that we do not have access to class labels to establish a supervised objective. Instead, we propose two novel self-supervised objectives: **temporal distance classification** (TDC), described in Section 3.1 and **cross-modal temporal distance classification** (CMC), described in Section 3.2. We also propose **cycle-consistency** in Section 3.3 as a quantitative measure for evaluating the one-to-one alignment capacity of an embedding.

### 3.1 Temporal distance classification (TDC)

We first consider the unsupervised task of predicting the temporal distance $\Delta t$ between two frames of a single video sequence. This task requires an understanding of how visual features move and transform over time, thus encouraging an embedding that learns meaningful abstractions of environment dynamics conditioned on agent interactions.

We cast this problem as a classification task, with $K$ categories corresponding to temporal distance intervals, $d_k \in \{[0], [1], [2], [3-4], [5-20], [21-200]\}$. Given two frames from the same video, $v, w \in I$, we learn to predict the interval $d_k$ s.t. $\Delta t \in d_k$. Specifically, we implement two functions: an embedding function $\phi : I \rightarrow \mathcal{R}^N$, and a classifier $\tau_{tdc} : \mathcal{R}^N \times \mathcal{R}^N \rightarrow \mathcal{R}^K$, both implemented as neural networks (see Section 5 for implementation details). We can then train $\tau_{tdc}(\phi(v), \phi(w))$ to predict the distribution over class labels, $d_k$, using the following cross-entropy classification loss:

$$L_{tdc}(v, w, y) = -\sum_{j=1}^{K} y_j \log(\hat{y}_j) \qquad \text{with } \hat{y} = \tau_{tdc}(\phi(v), \phi(w)), \qquad (1)$$

where $y$ and $\hat{y}$ are the true and predicted label distributions respectively.

### 3.2 Cross-modal temporal distance classification (CMC)

In addition to visual observations, our YouTube videos contain audio tracks that can be used to define an additional self-supervised task. As the audio of Atari games tends to correspond with salient events such as jumping, obtaining items or collecting points, a network that learns to correlate audio and visual observations should learn an abstraction that emphasizes important game events.

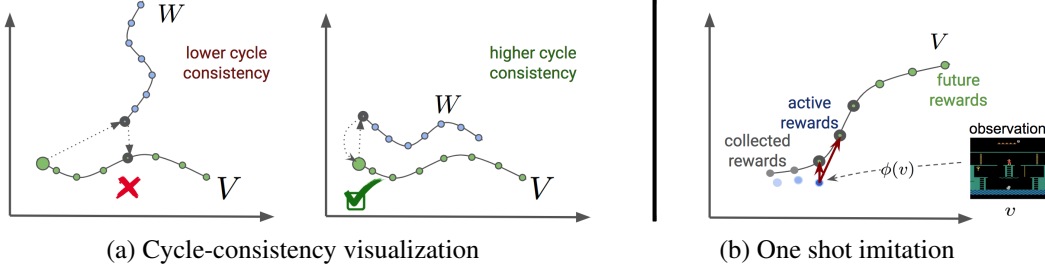

(a) Cycle-consistency visualization          (b) One shot imitation

Figure 4: (a) Visualization of two embedding spaces with low and high cycle-consistency. Note that the selected point in sequence $V$ (left) fails and (right) succeeds at cycling back to the original point. (b) Demonstration of one shot imitation through RL visualized in the embedding space.

We define the cross-modal classification task of predicting the temporal distance between a given video frame, $v \in I$, and audio snippet, $a \in A$. To achieve this, we introduce an additional embedding function, $\psi : A \to \mathcal{R}^N$, which maps from a frequency-decomposed audio snippet to an $N$-dimensional embedding vector, $\psi(a)$. The associated classification loss, $L_{cmc}(v, a, y)$, is equivalent to Equation (1) using the classification function $\hat{y} = \tau_{cmc}(\phi(v), \psi(a))$. Note that by limiting our categories to the two intervals $d_0 = [0]$ and $d_1 = [l, \ldots, \infty]$ with $l$ being the local positive neighborhood, this method reduces to $L^3$-Net of Arandjelovic et al. [3]. In our following experiments, we obtain a final embedding by a $\lambda$-weighted combination of both cross-modal and temporal distance classification losses, i.e. minimizing $L = L_{tdc} + \lambda L_{cmc}$.

### 3.3 Model selection through cycle-consistency

A challenge of evaluating and meta-optimizing the models presented in Section 3 is defining a measure of the quality of an embedding $\phi$. Motivated by the success of cyclic relations in CycleGAN [48] and for matching visual features across images [47], we propose cycle-consistency for this purpose. Assume that we have two length-$N$ sequences, $V = \{v_1, v_2, \ldots v_n\}$ and $W = \{w_1, w_2, \ldots, w_n\}$. We also define the distance, $d_\phi$, as the Euclidean distance in the associated embedding space, $d_\phi(v_i, w_j) = ||\phi(v_i) - \phi(w_j)||_2$. To evaluate cycle-consistency, we first select $v_i \in V$ and determine its nearest neighbor, $w_j = \mathrm{argmin}_{w \in W}\, d_\phi(v_i, w)$. We then repeat the process to find the nearest neighbor of $w_j$, i.e. $v_k = \mathrm{argmin}_{v \in V}\, d_\phi(v, w_j)$. We say that $v_i$ is *cycle-consistent* if and only if $|i - k| \leq 1$, and further define the *one-to-one alignment capacity*, $P_\phi$, of the embedding space $\phi$ as the percentage of $v \in V$ that are cycle-consistent. Figure 4(a) illustrates cycle-consistency in two example embedding spaces. The same process can be extended to evaluate the 3-cycle-consistency of $\phi$, $P_\phi^3$, by requiring that $v_i$ remains cycle consistent along both paths $V \to W \to U \to V$ and $V \to U \to W \to V$, where $U$ is a third sequence.

## 4 One-shot imitation from YouTube footage

In Section 3, we learned to extract features from unlabeled and unaligned gameplay footage, and introduced a measure to evaluate the quality of the learnt embedding. In this section, we describe how these features can be exploited to learn to play games with very sparse rewards, such as the infamously difficult PITFALL! and MONTEZUMA'S REVENGE. Specifically, we demonstrate how a sequence of checkpoints placed along the embedding of a single YouTube video can be presented as a reward signal to a standard reinforcement learning agent (IMPALA for our experiments [14]), allowing successful one-shot imitation even in the complete absence of the environment rewards.

Taking a single YouTube gameplay video, we simply generate a sequence of "checkpoints" every $N = 16$ frames along the embedded trajectory. We can then represent the following reward:

$$r_{\mathrm{imitation}} = \begin{cases} 0.5 & \text{if } \bar{\phi}(v_{\mathrm{agent}}) \cdot \bar{\phi}(v_{\mathrm{checkpoint}}) > \alpha \\ 0.0 & \text{otherwise} \end{cases} \qquad (2)$$

where $\bar{\phi}(v)$ are the zero-centered and $l_2$-normalized embeddings of the agent and checkpoint observations. We also require that checkpoints be visited in soft-order, i.e. if the last collected checkpoint is at

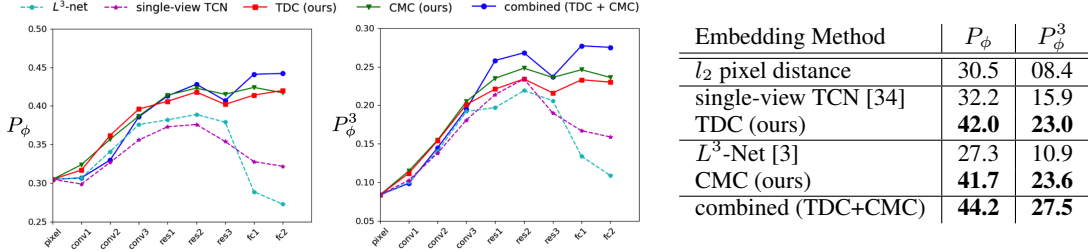

| Embedding Method | $P_\phi$ | $P_\phi^3$ |
|---|---|---|
| $l_2$ pixel distance | 30.5 | 08.4 |
| single-view TCN [34] | 32.2 | 15.9 |
| TDC (ours) | **42.0** | **23.0** |
| $L^3$-Net [3] | 27.3 | 10.9 |
| CMC (ours) | **41.7** | **23.6** |
| combined (TDC+CMC) | **44.2** | **27.5** |

Figure 5: Cycle-consistency evaluation considering different embedding spaces. We compare naive $l_2$ pixel loss to temporal methods (TDC and single-view TCN) and cross-modal methods (CMC and $L^3$-Net). Combining TDC and CMC yields the best performance for both 2 and 3-cycle-consistency, particularly at deeper levels of abstraction (e.g. no performance loss using FC1 or FC2).

$v^{(n)}$, then $v_{\text{checkpoint}} \in \{v^{(n+1)}, \ldots, v^{(n+1+\Delta t)}\}$. We set $\Delta t = 1$ and $\alpha = 0.5$ for our experiments (except when considering pixel-only embeddings, where $\alpha = 0.92$ provided the best performance).

## 5   Implementation Details

The visual embedding function, $\phi$, is composed of three spatial, padded, 3x3 convolutional layers with (32, 64, 64) channels and 2x2 max-pooling, followed by three residual-connected blocks with 64 channels and no down-sampling. Each layer is ReLU-activated and batch-normalized, and the output fed into a 2-layer 1024-wide MLP. The network input is a 128x128x3x4 tensor constructed by random spatial cropping of a stack of four consecutive 140x140 RGB images, sampled from our dataset. The final embedding vector is $l_2$-normalized.

The audio embedding function, $\psi$, is as per $\phi$ except that it has four, width-8, 1D convolutional layers with (32, 64, 128, 256) channels and 2x max-pooling, and a single width-1024 linear layer. The input is a width-137 (6ms) sample of 256 frequency channels, calculated using STFT. ReLU-activation and batch-normalization are applied throughout and the embedding vector is $l_2$-normalized.

The same shallow network architecture, $\tau$, is used for both temporal and cross-modal classification. Both input vectors are combined by element-wise multiplication, with the result fed into a 2-layer MLP with widths (1024, 6) and ReLU non-linearity in between. A visualization of these networks and their interaction is provided in Figure 3. Note that although $\tau_{tdc}$ and $\tau_{cmc}$ share the same architecture, they are operating on two different problems and therefore maintain separate sets of weights.

To generate training data, we sample input pairs $(v^i, w^i)$ (where $v^i$ and $w^i$ are sampled from the same domain) as follows. First, we sample a demonstration sequence from our three training videos. Next, we sample both an interval, $d_k \in \{[0], [1], [2], [3-4], [5-20], [21-200]\}$, and a distance, $\Delta t \in d_k$. Finally, we randomly select a pair of frames from the sequence with temporal distance $\Delta t$. The model is trained with Adam using a learning rate of $10^{-4}$ and batch size of 32 for 200,000 steps.

As described in Section 4, our imitation loss is constructed by generating checkpoints every $N = 16$ frames along the $\phi$-embedded observation sequence of a single YouTube video. We train an agent using the sum of imitation and (optionally) environment rewards. We use the distributed A3C RL agent IMPALA [14] with 100 actors for our experiments. The only modification we make to the published network is to calculate the distance (as per Equation(2)) between the agent and the next two checkpoints and concatenate this 2-vector with the flattened output of the last convolutional layer. We also tried re-starting our agent from checkpoints recorded along its trajectory, similar to Hosu et al. [23], but found that it provided minimal improvement given even our very long demonstrations.

## 6   Analysis and Experiments

In this section we analyze (a) the learnt embedding spaces, and (b) the performance of our RL agent. We consider three Atari 2600 games that are considered very difficult exploration challenges: MON-TEZUMA'S REVENGE, PITFALL! and PRIVATE EYE. For each, we select four YouTube videos (three training and one test) of human gameplay, varying in duration from 3-to-10 minutes. Importantly,

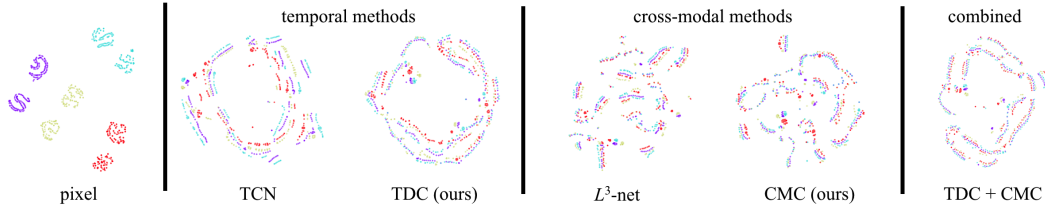

temporal methods       cross-modal methods       combined

pixel       TCN       TDC (ours)       $L^3$-net       CMC (ours)       TDC + CMC

Figure 6: For each embedding method, we visualize the t-SNE projection of four observation sequences traversing the first room of MONTEZUMA'S REVENGE. Using pixel space alone fails to provide any meaningful cross-domain alignment. Purely cross-modal methods perform better, but produce a very scattered embedding due to missing long-range dependencies. The combination of temporal and cross-modal objectives yields the best alignment and continuity of trajectories.

none of the YouTube videos were collected using our specific Arcade Learning Environment [10], and the only pre-processing that we apply is keypoint-based (i.e. Harris corners) affine transformation to spatially align the game screens from the first frame only. The dataset used and additional experiments can be found in the supplemental material to this paper.

## 6.1 Embedding space evaluation

To usefully close the domain gap between YouTube gameplay footage and our environment observations, our learnt embedding space should exhibit two desirable properties: (1) one-to-one alignment capacity and (2) meaningful abstraction. We consider each of these properties in turn.

First, one-to-one alignment is desirable for reliably mapping observations between different sequences. We evaluate this property using the cycle-consistency measure introduced in Section 3.3. The features from earlier layers in $\phi$ (see Figure 5) are centered and $l_2$-normalized before computing cycle-consistency. Specifically, we consider both (a) the 2-way cycle-consistency, $P_\phi$, between the test video and the first training video, and (b) the 3-way cycle-consistency, $P_\phi^3$, between the test video and the first two training videos. These results are presented in Figure 5, comparing the cycle-consistencies of our TDC, CMC and combined methods to a naive $l_2$-distance in pixel space, single-view time-contrastive networks (TCN) [34] and $L^3$-Net [3]. Note that we implemented single-view TCN and $L^3$-Net in our framework and tuned the hyperparameters to achieve the best $P_\phi^3$ cycle-consistency. As expected, pixel loss performs worst in the presence of sequence-to-sequence domain gaps. Our TDC and CMC methods alone yield improved performance compared to TCN and $L^3$-Net (particularly at deeper levels of abstraction), and combining both methods provides the best results overall.

Next, Figure 6 shows the t-SNE projection of observation trajectories taken by different human demonstrators to traverse the first room of MONTEZUMA'S REVENGE. It is again evident that a pixel-based loss entirely fails to align the sequences. The embeddings learnt using purely cross-modal alignment (i.e. $L^3$-Net and CMC) perform better but still yield particularly scattered and disjoint trajectories, which is an undesirable property likely due to the sparsity of salient audio signals. TDC and our combined TDC+CMC methods provide the more globally consistent trajectories, and are less likely to produce false-positives with respect to the distance metric described in Section 4.

Finally, a useful embedding should provide a useful abstraction that encodes meaningful, high-level information of the game while ignoring irrelevant features. To aid in visualizing this property, Figure 7 demonstrates the spatial activation of neurons in the final convolutional layer of the embedding network $\phi$, using the visualization method proposed in [46]. It is compelling that the top activations are centered on features including the player and enemy positions in addition to the inventory state, which is informative of the next location that needs to be explored (e.g. if we have collected the key required to open a door). Important objects such as the key are emphasized more in the cross-modal and combined embeddings, likely due to the unique sounds that are played when collected (see figure 7(d) and (e)). Notably absent are activations associated with distractors such as the moving sand animation, or video-specific artifacts indicative of the domain gap we wished to close.

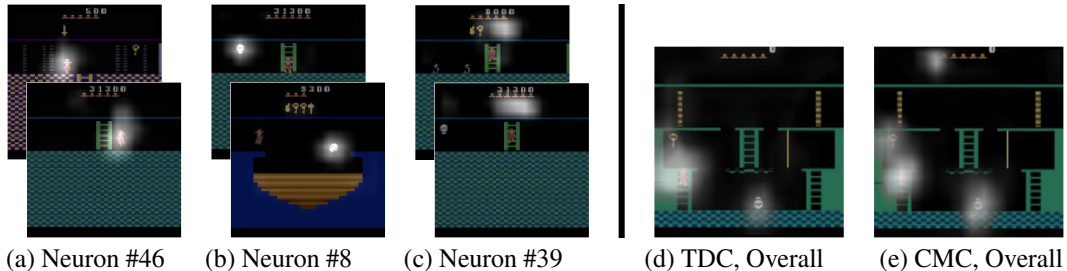

(a) Neuron #46     (b) Neuron #8     (c) Neuron #39     (d) TDC, Overall     (e) CMC, Overall

Figure 7: (Left) Visualization of select activations in the final convolutional layer. Individual neurons focus on e.g. (a) the player, (b) enemies, and (c) the inventory. Notably absent are activations associated with distractors or domain-specific artifacts. (Right) Visualization of activations summed across all channels in the final layer. We observe that use of the audio signal in CMC results in more emphasis being placed on key items and their location in the inventory.

## 6.2 Solving hard exploration games with one-shot imitation

Using the method described in Section 4, we train an IMPALA agent to play the hard exploration Atari games MONTEZUMA'S REVENGE, PITFALL! and PRIVATE EYE using a learned auxiliary reward to guide exploration. For each game, the embedding network, $\phi$, was trained using just three YouTube videos, and an additional video was embedded to generate a sequence of exploration checkpoints. Videos of our agent playing these games can be found here[2].

Figure 8 presents our learning curves for each hard exploration Atari game. Without imitation reward, the pure RL agent is unable to collect any of the sparse rewards in MONTEZUMA'S REVENGE and PITFALL!, and only reaches the first two rewards in PRIVATE EYE (consistent with previous studies using DQN variants [19, 22]). Using pixel-space features, the guided agent is able to obtain 17k points in PRIVATE EYE but still fails to make progress in the other games. Replacing a pixel embedding with our combined TDC+CMC embedding convincingly yields the best results, even if the agent is presented only with our TDC+CMC imitation reward (i.e. no environment reward).

To test the impact of the choice of expert trajectory, we generate checkpoints from two additional videos of MONTEZUMA'S REVENGE from our set, and train agents with those sequences (figure 8, left). While all three agents manage to clear the first level, expert 1 achieves the highest score. Out of the three expert sequence considered, expert 1 also has the biggest domain shift. This is in line with our findings from section 6.1 that our embedding space can sufficiently align our sequences. Domain shift in the expert trajectories is therefore not a significant factor on performance.

Finally, in Table 1 we compare our best policies for each game to the best previously published results; Rainbow [19] and ApeX DQN [22] without demonstrations, and DQfD [20] using expert demonstrations. Unlike DQfD our demonstrations are unaligned YouTube footage without access to action or reward trajectories. Our results are calculated using the standard approach of averaging over 200 episodes initialized using a random 1-to-30 no-op actions. Importantly, our approach is the first to convincingly exceed human-level performance on all three games – even in the absence of an environment reward signal. We are the first to solve the entire first level of MONTEZUMA'S REVENGE and PRIVATE EYE, and substantially outperform state-of-the-art on PITFALL!.

## 7  Conclusion

In this paper, we propose a method of guiding agent exploration through hard exploration challenges by watching YouTube videos. Unlike traditional methods of imitation learning, where demonstrations are generated under controlled conditions with access to action and reward sequences, YouTube videos contain only unaligned and often noisy audio-visual sequences. We have proposed novel self-supervised objectives that allow a domain-invariant representation to be learnt across videos, and described a one-shot imitation mechanism for guiding agent exploration by embedding checkpoints throughout this space. Combining these methods with a standard IMPALA agent, we demonstrate

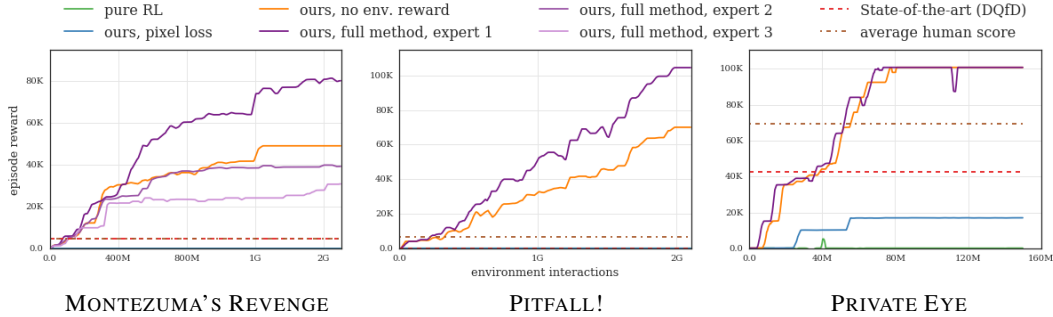

Figure 8: Learning curves of our combined TDC+CMC algorithm with (purple) and without (yellow) environment reward, versus imitation from pixel-space features (blue) and IMPALA without demonstrations (green). The red line represents the maximum reward achieved using previously published methods, and the brown line denotes the score obtained by an average human player.

|  | MONTEZUMA'S REVENGE | PITFALL! | PRIVATE EYE |
|---|---|---|---|
| Rainbow [19] | 384.0 | 0.0 | 4,234.0 |
| ApeX [22] | 2,500.0 | -0.6 | 49.8 |
| DQfD [20] | 4,659.7 | 57.3 | 42,457.2 |
| Average Human [43] | 4,743.0 | 6,464.0 | 69,571.0 |
| Ours ($r_{\text{imitation}}$ only) | 37,232.7 | 54,912.4 | 98,212.5 |
| Ours ($r_{\text{imitation}} + r_{\text{env}}$) | **58,175.1** | **76,812.5** | **98,763.2** |

Table 1: Comparison of our best policy (mean of 200 evaluation episodes) to previously published results across MONTEZUMA'S REVENGE, PITFALL! and PRIVATE EYE. Our agent is the first to exceed average human-level performance on all three games, even without environment rewards.

the first human-level performance in the infamously difficult exploration games MONTEZUMA'S REVENGE, PITFALL! and PRIVATE EYE.

**Acknowledgments**    We would like to thank the team, especially Serkan Cabi, Bilal Piot and Tobias Pohlen, for many fruitful discussions. We thank the reviewers for their comments, which helped in making this a better paper. And finally, we say 'thank you' to all the amazing Atari players on Youtube and Twitch, which inspired this project.

## Footnotes

[2]https://www.youtube.com/playlist?list=PLZuOGGtntKlaOoq_8wk5aKgE_u_Qcpqhu

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
