[Supplementary Material]



Figure 1: First frames of all videos in our dataset, after pre-processing.

# 1 Dataset

For each Atari game we consider, we chose four game-play videos on Youtube. We aimed at selecting videos with different visual appearance, to make our learnt embedding robust against distractors. MONTEZUMA'S REVENGE contains an additional validation video far outside the distribution, which we only used to test the robustness of the trained embedding in aligning sequences.

Fig. 1 shows the first frames of these videos after pre-processing, i.e., clipping the image with the coordinates listed below, and re-sizing to $140 \times 140$ pixels. We start the video at the time indicated in the table below. For training the embedding, we limit video duration to 9000 frames(10 mins); for creating the checkpoints, we use the whole duration of the video.

| Game | Video link https://youtu.be/... | high score | clip window $(x_1, y_1, x_2, y_2)$ | start time |
|---|---|---|---|---|
| MONTEZUMA'S REVENGE | sYbBgkP9aMo | 242,900 | 0, 22, 480, 341 | 0.0s |
| | 6zXXZvVvTFs | 54,700 | 35, 50, 445, 300 | 0.6s |
| | SuZVyOlgVek | 72,700 | 79, 18, 560, 360 | 0.2s |
| | 2AYaxTiWKoY | 6,100 | 0, 13, 640, 335 | 8.8s |
| | pF6xCZA72o0 | 11,900 | 20, 3, 620, 360 | 24.1s |
| PRIVATE EYE | zfdov0gmPRM | 101,800 | 382, 37, 1221, 681 | 14.5s |
| | YvaSsTIbvfc | 101,600 | 235, 64, 1043, 654 | 7.6s |
| | 3Rqxqrbi9KE | 101,500 | -59, -18, 1128, 765 | 2.8s |
| | jt0YhI_CYUs | 101,800 | 296, 0, 1280, 718 | 4.3s |
| PITFALL! | CkDllyETiBA | 113,332 | 123, 26, 527, 341 | 25.6s |
| | aAJzamWAFOE | 64,275 | -30, -11, 570, 381 | 8.3s |
| | sH4UpYOeDFA | 27,334 | 0, 44, 480, 359 | 8.3s |
| | BPYsqj5N_0I | 114,000 | 70, 21, 506, 359 | 1.0s |

# 2 Game-play videos

You can observe our learnt agent playing Atari games in the following videos.

- PRIVATE EYE: https://youtu.be/I5itifmdrEo

- PITFALL!: https://youtu.be/3XMJR7z2KMc

- MONTEZUMA'S REVENGE: https://youtu.be/luDm-ieOOmA

Figure 2: Learning curves of our algorithm on MONTEZUMA'S REVENGE, when trained end-to-end (purple), from pre-trained TDC+CMC features with (green) and without (yellow) conditioning on the features of the next checkpoint.

In https://youtu.be/DhUqfLjvONY, we demonstrate how our agent interprets expert videos with the example of MONTEZUMA'S REVENGE. Notice that our agent learns to optimize the timing of jumps over the enemy, while still following the expert's trajectory.

## 3  Learning from features

If pre-trained TDC+CMC features work well as a similarity metric, we can consider to also learn the policy itself from features. Learning in feature space enables significant reductions in the size of the policy network. It is also very attractive for off-policy learning, as it allows us to store a small feature vector, instead of the full input image, in the replay table.

In an experiment, we replaced the convolutional layers of IMPALA's policy network with our frozen, pre-trained embedding network. The policy network therefore reduces to two linear layers of width 256, which take an embedding vector of dimension 1024 as input, and output policy and baseline. This reduces the number of trainable parameters by about 60%. The generation of imitation rewards, and all other aspects of the algorithm remain unchanged.

In figure 2 we compare the performance of the method when learning from features to the full method. While the performance does not quite match the original method, it still significantly outperforms the state-of-the-art. This is quite surprising, considering that the feature layers were pre-trained without any environment interactions, and may miss certain screen elements. As the inputs of our agent are now in feature space, we also can condition the agent on the the next checkpoint in sequence, simply by providing the checkpoints' embedding as an additional input. This allows the agent to compute a rich similarity signal, which significantly speeds up learning (see green curve in figure 2).

## 4  Visualization of the learnt embedding space

In figure 3, we visualize the t-SNE projection of four observation sequences using embeddings computed using TCN, TDC, $L^3$-Net, CMC and our combined method. The trajectories traverse the first room of MONTEZUMA'S REVENGE, i.e. moving from the start position to the key location, and back to the start position.

The feature vectors from earlier convolutional layers fail to provide any meaningful cross-domain alignment, while later layers can align the trajectories. In the later layers, we observe a strong difference in continuity between the embedding methods considered. The embeddings learnt using purely cross-modal alignment, i.e. $L^3$-Net and CMC, yield particularly scattered and disjoint trajectories. On the other hand, TCN reduces the state space to a single curve, despite the fact that we should see some similarity between the first half (start to key) and the second half (key back to start) of the trajectory. The combination of temporal and cross-modal objectives yields very good alignment and continuity of trajectories.

The video https://youtu.be/RyxPAYhQ-Vo shows an animation of four expert videos, temporally aligned using our combined embedding. Note that while the four observation sequences are mostly traversing the same path, there are small differences due to e.g. different timing of jumps, which can be observed as divergences in the embedding space.

Figure 3: Visualization of the t-SNE projection of four observation sequences traversing the first room of MONTEZUMA'S REVENGE.