[Reviews · NeurIPS 2018]

Reviewer 1



This submission proposes a method to learn dense reward functions for hard exploration games with sparse rewards using videos of humans playing the game from Youtube. The domain gap between YouTube videos and the simulator is closed using a set of novel self-supervised classification tasks such as temporal distance classification and cross-modal distance classification to learn a common representation. The reward function learned by the proposed method can be used to train RL agents which receive much higher rewards on 3 hard exploration games as compared to agents trained with the sparse environment reward. Strengths: - The paper is very well written and easy to follow. - The problem setup has a lot of practical applications. There are plenty of videos available on the web for a lot of useful tasks. Learning from videos is much more useful than learning from data recorded directly from simulators under artificial conditions. The method doesn't require frame-by-frame alignment or actions at each observation. - The method achieves super-human performance in 3 hard exploration Atari games. - The method outperforms several strong baselines. - The method utilizes the audio channel in videos in an interesting and effective manner. - The visualizations of both embedding spaces and convolutional activations provide informative qualitative analysis. Questions - The authors select 4 YouTube videos of human gameplay for each game. How were these videos selected? Randomly or based on performance, visual similarity, temporal alignment or a combination of these? - What was the average performance of these 4 videos? What was the performance of the video used to learn the imitation reward? - Is the implementation code going to be open-sourced? UPDATE: The authors have answered my questions about the scores and selection of the demonstration videos in the response. I encourage the authors to add these details to the manuscript and open-source their implementation and dataset.

Reviewer 2



Summary: This paper studies learning from demonstration/imitation learning. While usually supervision has to be intentionally provided by a human, the authors instead use YouTube videos as a form of supervision. They first align videos to a shared representation, using two concept that do not require any labeling: i) predicting the temporal distance between frames in the same video, and ii) predicting the temporal distance between a video and audio frame of the same video. Subsequently, they use the embedding on a novel video to generate checkpoints, which serve a intermediate rewards for an RL agent. Experiments show state-of-the-art performance amongst learning from demonstration approaches. Strength: - The paper is clearly written, with logical steps between the sections, and good motivations. - Related work is well covered. - The results are impressive, and exceed the state-of-the-art on the studied games. Weakness/Questions: - Introduction: You mention intrinsic motivation, but you miss another branch of exploration work, based on uncertain value functions. See for example [1,2]. -In the beginning of Section 4 I did not understand why you could suddenly sample checkpoints, while in Sec. 3 you only focused on learning a frame-wise embedding. The end of Sec 5. does explain how this is implemented (with a held-out video), which could be mentioned a bit earlier for clarity. - How much did you select the YouTube demonstrations? Were they expert players, who were very good at solving these games? That might explain why you results are better than e.g. DqfD? - Atari games have a fixed viewpoint, and you additionally spatially align the demonstrations if the videos don’t. To what extend is spatial alignment crucial? Would this approach scale to first-person-view games, where alignment is probably much harder? Conclusion: I believe this paper studies an interesting problem. There is big potential for the use of (human) supervision to guide learning, but supervision is a time consuming process. This paper studies video data as a potential form of supervision, which would indeed provide access to a huge amount of demonstrations. The paper contributes a novel method for alignment, and a simple but effective way to incorporate this embedding in an RL agent, which shows good results. It would be good to mention how expert the demonstration videos were, and maybe discuss the potential to generalize to non-fixed viewpoints. Besides that, I think this paper makes a clear contribution. [1] White, D. J. "Uncertain value functions." Management Science 19.1 (1972): 31-41. [2] Osband, Ian, et al. "Deep exploration via randomized value functions." arXiv preprint arXiv:1703.07608 (2017). After rebuttal: It is important to mention the scores and acquisition procedure of the demonstration videos indeed. They were expert demonstrations, but as you mention yourself, a benefit of your approach is that (through the alignment) you can leverage a series of unlabeled expert demonstrations.

Reviewer 3



This paper aims at leveraging human demonstration to improve exploration in a deep RL setting. In particular, this paper deals with the problem of overcoming domain gaps in the imitation learning setting and without access to the actions and rewards that led to the demonstrator’s observation trajectory. The paper is well-written and provides interesting results. Equation (2) defines a positive reward when the zero-centered and l2-normalized embeddings of the agent is similar to "checkpoints" of a single youtube video. In that case, the agent may want to repeatedly stay in the same set of states to repeatedly gather the reward and may be blocked in a local optimum. I suppose that this is why the checkpoints should be visited in a "soft order". It would be interesting to clearly state this. In the experiments, four YouTube videos of human gameplay are used. It is not explicitly stated whether the domain gap of these four youtube videos are the ones illustrated in Figure 1. A good addition to the paper would be to provide for all three games (MONTEZUMA’S REVENGE, PITFALL! and PRIVATE EYE) an illustration of the frames after pre-processing with keypoint-based (i.e. Harris corners) affine transformation so that one can visualize the domain gap. In addition, it could be a good addition to provide a visual illustration of the held-aside video so that the reader can understand the domain gap closed between the video used to generate the "checkpoints" and the one from ALE. It would also be interesting to show that the algorithm is robust when using another video from the set of four. Indeed, the ALE frame is close to some of the frames from the youtube video in Figure 1b and it would be possible that the algorithm is particularly efficient for some of them but not all. Additional remarks: - typo: line 82 "the the". - In equation 2, it may be wiser to choose a different sign than \gamma, because that one is usually used for the discount factor in RL.